# Effects of Castration Age on the Growth Performance of Nubian Crossbred Male Goats

**DOI:** 10.3390/ani12243516

**Published:** 2022-12-13

**Authors:** Yu-An Chen, Jing-Yan Chen, Wei-Qun Chen, Wen-Yen Wang, Hsi-Hsun Wu

**Affiliations:** Department of Animal Sciences, National Pingtung University of Science and Technology, 1, Shuefu Road, Neipu, Pingtung 91201, Taiwan

**Keywords:** castration, Nubian crossbred goats, growth performance, ultrasonic measurement

## Abstract

**Simple Summary:**

Castration is a common management practice used to improve the meat quality of goats, but the optimal timing to perform castration has yet to be scientifically determined. As a result, goat breeders in Taiwan have discrepant practices regarding the timing of castration. Some producers believe that late castration results in better carcass performance, whereas others claim that earlier castration minimizes the stress associated with the operation. The purpose of this study was to compare the growth performance and economic costs between goats castrated at 3 months (weaning) and 6 months (onset of puberty) of age, respectively. Results from this study provide fundamental information for goat farmers to evaluate the effects of castration timing on various production-related parameters, such as growth performance, carcass quality, and negative impacts caused by late castration.

**Abstract:**

To determine the optimal timing for performing castration on goats, eighteen male Nubian crossbred goats were randomly assigned to two groups and castrated at 3 months and 6 months of age, respectively. Daily dry matter intake, biweekly body weights, and ultrasonic measurements of *longissimus dorsi* muscle growth were recorded. Results indicated that there was no significant difference between the two groups in terms of the blood parameter analysis (except testosterone, 0.36 ± 0.26 vs. 3.61 ± 0.27 ng/mL at 25 weeks old), economic analysis, and growth performance, including final body weight, total weight gain, average daily gain, total dry-matter intake, and feed conversion ratio (*p* > 0.05). However, the *longissimus dorsi* muscle depth of goats castrated at 6 months of age was significantly higher than that of goats castrated at 3 months of age. In conclusion, castration timing does not have a significant effect on the growth performance of goats; therefore, castrating goats at 3 months of age may be the best practice considering animal welfare and possible risks associated with late castration.

## 1. Introduction

The castration of animals for meat production is a standard management practice used to improve meat quality, render male animals more manageable, avoid unwanted pregnancies, and prevent the propagation of heritable defects [1]. In male goats, the peculiar odor is produced in sebaceous glands by testosterone [2], where branched-chain fatty acids in the serum are identified as the odor components and then transferred from the blood to the meat. Castrating male goats in the late growth stage was believed to result in better muscular growth based on the fact that mature body weight and size in the castrated male goat became smaller than that of the intact male goat due to the increased secretion of female hormones such as the follicle-stimulating hormone and luteinizing hormone comparing to intact male goats [3,4]. Therefore, it is possible to improve meat quality by castration; for example, the marbling score of the meat is improved and the peculiar odor is reduced [5]. According to Zeng et al., known castration techniques include surgical castration, mechanical castration, chemical castration, and immunocastration [6]. From the survey by Hempstead et al., castration was mostly carried out with a ring or band (mechanical castration) because minimal training is required; surgical castration does require techniques and this method can also cause acute and long-term pain [7].

Nevertheless, the effect of castration on the growth performance of goats and carcass characteristics is equivocal [8]. For instance, Damascus male kids recorded higher daily weight gains than females and early castrated male kids that were not yet 7.5 months of age, the average daily gain (ADG) being adversely affected by castration [9]. On the other hand, Koyuncu [10] reported that castration of hair goat kids in Turkey at 100 days of age did not have a significant effect on the ADG between 160 and 216 days of age. As for carcass characteristics, the castration of goats generally resulted in an increase in fat deposition in the body [10,11,12]. In contrast, no significant effect of castration on fat deposition was observed in goats from the study of Tahir [13].

A limited number of studies address the effect of castration age on the growth and the feeding cost of goats. In Taiwan, most goat breeders believed that castrating goats at 6 months of age (the fattening period) allows androgen to stimulate more muscle growth [14] and selected for further breeding bucks. However, some commercial goat farms castrated goats at 2–3 months of age for more practical reasons, such as lowering the risk of prolonged recovery of the wound [15], having a much greater effect on marbling degree, and preventing unwanted mating. Therefore, the objective of this study was to investigate the effect of performing castration at 3 months and 6 months of age of in Nubian crossbred goats on the growth performance, blood parameters, and economic analysis to provide insights into promoting efficient farm management as well as goat meat quality.

## 2. Materials and Methods

### 2.1. Animals and Experimental Design

#### 2.1.1. Animal Management

This research was approved by the National Pingtung University of Science and Technology—Institutional Animal Care and Use Committees (IACUC) (NPUST-107-008). Weaner Nubian × Boer crossbred male goats (*n* = 18) were purchased from a commercial meat goat farm at 2.5 months of age (15.15 ± 0.51 kg on average). The animals were reared in the experimental ranch on campus farm and randomly assigned into two groups according to their body weights. Three goats were housed in a pen (4 × 1.3 m^2^) during the experimental period. In the experiment, the growth duration was 169 days (from 16 April 2018 to 1 October 2018, BW from 15 kg to 40 kg) and the fattening period lasted for 97 days (from 22 October 2018 to 7 January 2019, BW from 40 kg to finish). The operations were performed on 16 April for the 3-month castration group and on 9 July for the 6-month castration group.

During the growth period, the crude protein, metabolizable energy content, and estimated dry matter intake of the diets were 13.84%, 2.58 Mcal/kg DM, and 3.46% of BW, respectively, which meet the nutrient requirements as per NRC [16] for targeting body weight gain to be 200g/day. For the fattening period, the crude protein, metabolizable energy content, and estimated dry matter intake of the diet were 13.63%, 2.48 Mcal/kg DM, and 2.91% of BW, respectively, and according to the nutrient requirements per NRC [16], the target weight gain was expected to be 200 g/day (Table 1). The diets were equally divided into two parts, with feeding at 6:30 a.m. and 6:00 p.m. every day, and freshwater was provided ad libitum. Pangola hay and alfalfa pellets were fed first, and a mixture of concentrate, HAS (commercial bypass starch product), and bypass fat powder was given 30 min after the feeding forage. The residue feed was collected at 5:30 p.m. and the next day at 6:00 a.m., weighed, and recorded for calculating feed intake.

The goats were weighed biweekly to calculate the ADG. In addition, the increase of diets provided for each group was adjusted according to the NRC [16] nutritional requirements and the weight variation of the goat.

#### 2.1.2. Castration

In this trial, the goats were castrated by surgical operation, which is more suitable for young animals, we effectively reached the intended goal assuming the operation was performed appropriately and post-surgery treatments were under supervision to preventing infections [6]. All the facilities, hands, handling, and instruments were disinfected with iodine. First, goats were sedated with intramuscular xylazine (0.1 mg/kg; Balanzine 2%^®^, 20 mg/mL, EBS, Taipei, Taiwan), and anesthesia was induced with intravenous Zoletil (1 mg/kg; Zoletil^®^, 50 mg/mL, Virbac, Taipei, Taiwan). Then, the castration was carried out by trained personnel under the supervision of a veterinarian. After the surgery, Penimycin (0.02~0.04 mL/kg; Penimycin-S^®^, CCPG, Tainan, Taiwan) and iodine were injected into the scrotum for sterilization. Sulpyrin (2~8 mL for goats; Sulpyrin 50%^®^, 500 mg/mL, CCPG, Tainan, Taiwan) was injected intramuscularly at the thigh to prevent pain and inflammation. After the operation, the goats were placed in a clean, wide, and ventilated pen with proper wound care and intensive checkup for faster recovery.

#### 2.1.3. Vaccination

The goats were vaccinated with the goat pox and foot-and-mouth disease (FMD) vaccines before the trial was initiated. To ensure animal health during the trial, 0.5% of Ivermectin (the recommended dosage of the product is 0.1 mL/kg) was drenched from the bulge of the scapula to the tail for deworming, and 20% of sulfanilamide (the recommended dosage of the product was 0.06–0.1 mL/kg) was injected intramuscularly for endoparasites.

### 2.2. Measurements and Analysis

#### 2.2.1. Ultrasonic Measurements

The *longissimus dorsi* muscle area (LMA), depth (LMD), and width (LMW) were measured biweekly during the experiment. According to Yarali [17] and Sahin [18], the area between the 12th and 13th ribs on the left side of the goat was used as ultrasonic measurement site. The combination of ultrasound coupling gel and salad oil was used to display the image more clearly. The probe and the rib were maintained parallel. The images captured from the ultrasonic instrument (EXAGO, ECM) were manually delineated and measured for area (cm^2^), depth (mm), and width (mm) of muscle with a built-in electronic caliper. All of the measurements were operated by the same personnel. The measurement time was synchronized with the weighing time (from 9:00 to 11:00).

#### 2.2.2. Blood Biochemical Analysis

For comparison, blood was collected at three stages: before the start of castration at 3 months of age and at 6 months of age and at the end of the fattening period (on 16 December 2018). The goats were restrained with hands, and blood samples were collected through the jugular vein. The vacutainers were kept with ice bath for 2 h to allow the blood to coagulate and then centrifugated (CN-3100, Hsiangtai) with a built-in radius of gyration of 7.5 cm (RS-50, Hsiangtai) at 3000 rpm for 10 min. Then, the supernatant was collected and stored in a freezer at −20 °C for further analysis.

The serum biochemical components estimated were total protein (TP), globulin (GLB), albumin (ALB), serum urea nitrogen (SUN), triglyceride (TG), cholesterol (CHOL), and blood sugar (glucose, Glu); while the serum enzyme activity measured were aspartate aminotransferase (AST), alanine aminotransferase (ALT), alkaline phosphatase (ALK-P); and androgen measurement was done through testosterone (TES). The samples were measured by commercial kits on the Roche Modular Analytics P800 (Roche Diagnostics, Indianapolis, IN, USA).

### 2.3. Statistical Analysis

Statistical analysis was performed using the Statistical Analysis Software Package version 9.4 (SAS Inst. Inc., Cary, NC, USA) and variance analysis was performed using the general linear model procedure (GLM). The significance of differences found between the treatments was compared using the least-square means (LSM).

## 3. Results and Discussion

### 3.1. Effect of Different Castration Timing on Growth Performance

The growth performance, dry mater intake, feed conversion ratio, and feed cost per kg weight gain were not significantly (*p* > 0.05) different between goats castrated at 3 months and 6 months of age from the growing to fattening period (Table 2), which was consistent with the results of Kebede [14] indicating that there was no significant effect of Arsi-Bale goats castrated at 3, 6, and 9 months old on total weight gain and ADG. Likewise, Yacob indicated that castration age had no significant effect on body weight or ADG of Black-Head Somali rams [19]; however, the depression of growth performance can be observed on the goat immediately fattening for sale after castration [20]. The results of Zamiri et al. demonstrated that castrating local goats at 3 months of age leads to a lower ADG after one month of fattening, the depression of which was postulated to be due to the castration procedure using Burdizzo method [21]. Our results indicated that the goats with enough recovery time after castration (fattening for 3 months) showed no significant difference between the two castration timings in terms of growth performance, which was consistent with the recommended timing from Zamiri et al. [21]. When it comes to efficiency, at the fattening period, castrating at 3 months of age showed higher efficiency than at 6 months of age regarding feed cost per kg weight gain (77.66 vs. 86.52). Yet, the fact that the feed conversion ratio at the growing and fattening stage were not significantly different may be due to the high adaptability of Nubian crossbred goats. Besides, there was one goat castrated at 6 months of age that was stressed by the operation, and a larger body conformation is a difficult constraint on animals, so the goat died the next day after castration, causing significant economic loss. As a result, other factors, such as the ease of management and the stress level of animals could be considered when deciding the castration timing for goats, since the castration carried out in younger animals can reduce the chance of complications [15].

### 3.2. Effects of Performing Castration at Different Ages on Longissimus Dorsi Muscle Growth

At the growing period, the Nubian crossbred goats castrated at 6 months of age showed a significantly higher (*p* < 0.05) LMD (Table 3), which may be caused by the testicular hormones resulting in a greater muscle growth capacity in intact males [22], yet, at the end of fattening period, there was no significant difference between two groups (Table 3). When it came to the growth of LMA, there was no significant difference between two groups, which was consistent with the result of Kebede [14] that castration age had no significant effect on the LMA. In support of this finding, Hopkins-Shoemaker [23] did not find a significant difference between the LMA of intact and castrated goats. Back-fat thickness in goats castrated at 3 months of age was lower than that of goats castrated at 6 months of age (*p* = 0.0962). Likewise, the results of Bayarktaroglu showed that the early castration of Saanen × Kilis goats significantly increased body fat deposition, except for the back-fat thickness. On the other hand, the results of Zamiri et al. indicated that the back-fat thickness growth of castrates with longer fattening period was generally greater than that of the intact males, yet no significant difference was found [21]. Therefore, we can conclude that the percentage of fat in the carcass of goats is affected by the castration as well as the duration of the fattening period, which may be a factor for farmers to consider in improving the goat-meat quality [5].

### 3.3. Effects of Performing Castration at Different Ages on Biochemistry Analysis of Blood

The blood parameters, including total protein, globulin, albumin, serum urea nitrogen, total cholesterol, and glucose, were in the normal range of concentration in intact goats and castrated goats during growing period (Table 4), and there was no significant difference observed between the two groups during fattening period (Table 5). It was noticeable that in the analysis of triglyceride (TG), the concentration in the growing period was higher than the normal value indicated in the literature, it was inferred that the goats had not reached the age of sexual maturation, so that the concentration of triglycerides in serum was relatively high. As the age increased, the concentration reduced. Other reasons may be responsible for the difference may include: the breed, environment, feeding management, and analysis method [24,25]. At the end of the fattening period, the value of TG was in the normal range, consistent with the biochemical values of goats described by Bogin [26]. Moreover, there was no significant difference in the concentration of TG between the two groups (Table 5). The serum-enzyme activity parameters, including AST and ALT, in intact and castrated goats, both fell within the normal range, according to AHDC [27], and were not affected by the timing of castration. However, increased activity of the ALK-p can be found in growing animals or animals with increasing osteoblast activity. In the trial, the value had dropped in the fattening period compared to the 25-week-old (Table 4), which corresponds to the result of Sharma [28] that the activity of the ALK-p-value was higher in the young or growth-period animals. The plasma concentration of TES was mainly controlled by the photoperiod, as well as genotype, age, and feeding level [29]. In the trial, the TES concentration was measured for goats of 6–7 months of age in the non-breeding season, and the result for the goats, whether castrated or not, fell in the normal range and was significantly higher in the non-castrated group (*p* < 0.01), which corresponded to a significant higher growth of LMD for the 6-month castration group in the growing period (Table 4).

To summarize, from the results of the biochemical analysis of blood parameters, there was no significant difference between intact and castrated goats, which indicates that, even castrated, goats were still in healthy condition, based on the blood parameter results. The only significant difference was found in the TES concentration (Table 4) since the castration was done in the non-breeding season and the goats had not yet reached puberty, so the TES concentration was low, which was consistent with the results of Souza et al. [29] and Theron [30].

**Table 4 animals-12-03516-t004:** Effects of castration and non-castration on serum blood biochemistry parameters of Nubian crossbred male goats at 25 weeks old (LSM ± SE).

Item	Castration	Non-Castration	Ref. ^1^
Aspartate aminotransferase, AST (U/L)	104.6 ± 7.71 (9)	82.9 ± 8.32 (8)	58~513
Alanine aminotransferase, ALT (U/L)	23.4 ± 1.72 (9)	21.3 ± 1.85 (8)	16~59
Alkaline phosphatase, ALK-p (g/dL)	373.0 ± 48.85 (8)	363.5 ± 52.76 (7)	97~387
Total protein, TP (g/dL)	6.6 ± 0.09 (8)	6.6 ± 0.10 (7)	6.2~8.0
Albumin, ALB (g/dL)	3.7 ± 0.06 (8)	3.6 ± 0.06 (8)	2.9~4.0
Globulin, GLB (g/dL)	2.9 ± 0.08 (9)	3.0 ± 0.09 (8)	2.7~4.1
Albumin/globulin ratio, A/G	1.3 ± 0.05 (8)	1.2 ± 0.05 (8)	1.0~2.0
Serum urea nitrogen, BUN (mg/dL)	17.7 ± 0.69 (7)	18.2 ± 0.74 (7)	9~35
Glucose PC, Glu-PC (mg/dL)	68.8 ± 2.23 (8)	67.1 ± 2.41 (8)	35~142
Triglyceride, TG (mg/dL)	41.8 ± 6.56 (8)	50.8 ± 7.09 (8)	10~29
Total Cholesterol, T-CHO (mg/dL)	85.8 ± 8.02 (7)	103.1 ± 8.66 (8)	73~280
Testosterone, TES (ng/mL)	0.36 ± 0.26 (3) ^b^	3.61 ± 0.27 (3) ^a^	2.6~14.2

^a, b^ Means within the same row without same letter on superscripts are significantly different (*p* < 0.05). ^1^ The reference values are ALT, AST, and T-CHO from AHDC [27], GLB, BUN, and ALK-p from Kaneko et al. [31]; TG from Cox et al. [32]; ALB, TP, Glu-PC, and A/G from AHDC [27]; TES of a 28-week-old Anglo Nubian goat from Souz et al. [29]. Value in parentheses is the number of goats.

**Table 5 animals-12-03516-t005:** Effects of castration at different ages on serum-blood biochemistry parameters in Nubian crossbred wethers at the end of the fattening period (LSM ± SE).

Item	Castration at 3-Month-Old	Castration at 6-Month-Old	Ref. ^1^
Aspartate aminotransferase, AST (U/L)	85.3 ± 2.80 (7)	79.8 ± 3.29 (6)	58~513
Alanine aminotransferase, ALT (U/L)	20.1 ± 0.65 (8)	19.4 ± 0.69 (7)	16~59
Alkaline phosphatase, ALK-p (g/dL)	257.2 ± 25.28 (8)	319.5 ± 28.66 (6)	97~387
Total protein, TP (g/dL)	6.4 ± 0.08 (8)	6.4 ± 0.10 (6)	6.2~8.0
Albumin, ALB (g/dL)	3.8 ± 0.07 (7)	3.9 ± 0.06 (8)	2.9~4.0
Globulin, GLB (g/dL)	2.6 ± 0.06 (9)	2.7 ± 0.08 (7)	2.7~4.1
Albumin/globulin ratio, A/G	1.4 ± 0.04 (8)	1.5 ± 0.05 (7)	1.0~2.0
Serum urea nitrogen, BUN (mg/dL)	14.5 ± 0.76 (7)	16.2 ± 0.76 (7)	9~35
Glucose PC, Glu-PC (mg/dL)	76.1 ± 2.05 (8)	73.9 ± 2.58 (6)	35~142
Triglyceride, TG (mg/dL)	22.6 ± 2.34 (7)	22.0 ± 2.34 (7)	10~29
Total Cholesterol, T-CHO (mg/dL)	111.4 ± 4.84 (7)	119.3 ± 5.93 (6)	73~280

^1^ The reference values are ALT, AST, and T-CHO from AHDC [27], GLB, BUN, and ALK-p from Kaneko [31]; TG from Cox [32]; ALB, TP, Glu-PC, and A/G from AHDC [27]; TES from Todini et al. [30]. Value in parentheses is the number of goats.

### 3.4. Effects of Perfoming Castration at Different Ages on Economic Benefits

The economic analysis of castration at different ages in Nubian crossbred male goats is shown in Table 6. Although the experimental feeding cost of the 3-month castration group was lower than that of the 6-month castration group (NT$ 3329 vs. NT$ 3537, respectively), the higher BW (58.02 vs. 59.79, respectively) generates higher net earnings for the 6-month castrated group than the 3-month castrated group (NT$ 2908 vs. NT$ 3358, respectively). If the experimental duration was prolonged until the goats reached the major marketing body weight (70 kg), the estimated cost would be feed cost per kg of weight gain in fattening period (77.66 vs. 86.52, Table 2), multiplying the weight gain from actual marketing body weight (58.02 vs. 59.79) to 70 kg; hence, the feed cost of the 3-month group would be NT$ 161 lower than that of the 6-month group (NT$ 4420 vs. NT$ 4259, respectively), and the net earnings would be higher (NT$ 5021 vs. NT$ 4860, respectively). Moreover, the risk of failure of castration could be withstood and the loss would be lower for the 3-month castration group. Therefore, for commercial breeders, the market factors according to their breeds, management method, growth, fattening, and marketing should be considered to maximize profits.

## 4. Conclusions

The age of castration for Nubian kids, either at 3 months or 6 months of age, did not have any significant difference on the growth performance, ultrasonic measurements of *longissimus dorsi* muscle growth, and the blood parameter analysis. However, the earlier the castration was implemented, the fewer negative impacts were observed since the operation caused larger wounds and a prolonged recovery period in larger animals. When animal welfare, growth performance, and feeding cost evaluations are taken into consideration, castrating goats at 3 months of age in Nubian crossbred goats is recommended.

## Figures and Tables

**Table 1 animals-12-03516-t001:** The ingredients and chemical composition of experimental diets.

Ingredients (as fed, %)	Growing Period	Fattening Period
Concentrate ^1^	28.27	26.73
Alfalfa pellet	23.39	21.62
HAS ^2^	27.72	32.43
Pangola hay	17.75	19.22
Energy booster 100^® 3^	2.88	-
Chemical composition	Growing period	Fattening period
Dry matter	89.02	88.93
Crude protein	13.84	13.63
Crude fat	6.46	3.79
Acid detergent fiber	20.11	20.23
Neutral detergent fiber	31.21	31.40
Ash	5.99	6.34
Metabolizable energy (Mcal/kg DM)	2.58	2.48

^1^ Concentrate (DM: 88.1%, CP: 19.6% DM, CF: 4.76% DM, ME: 2.65 Mcal/kg DM). ^2^ HAS: the bypass starch supplement purchased from Dachan Greatwall Group (DM: 88%, bypass starch: 35% DM, CP: 10.5% DM, CF: 5.4% DM, ME: 2.9 Mcal/kg DM). ^3^ Energy booster 100^®^: bypass fat (DM: 99.7%, CF: 99.00% DM, ME:8.25 Mcal/kg DM).

**Table 2 animals-12-03516-t002:** Effects of castration at different ages on the growth performance of Nubian crossbred male goats in the growing period and fattening period (LSM ± SE).

Items	Castration at 3-Month-Old (*n* = 9)	Castration at 6-Month Old (*n* = 8)	*p*-Value
Growing period ^1^			
Initial body weight (kg)	14.91 ± 0.50	15.39 ± 0.53	0.5227
Final body weight (kg)	39.28 ± 1.50	42.17 ± 1.25	0.1662
Total gain (kg)	24.37 ± 1.01	26.78 ± 1.07	0.1203
Average daily gain (g/d)	144.18 ± 5.95	158.47 ± 6.31	0.1203
Total dry matter intake (kg)	127.08 ± 12.54	138.17 ± 12.54	0.8219
Feed conversion ratio ^3^	5.22 ± 0.08	5.16 ± 0.08	0.4352
Feed cost per kg weight gain ^4^	76.88 ± 3.15	75.13 ± 3.16	-
Fattening period ^2^			
Initial body weight (kg)	39.28 ± 1.50	42.17 ± 1.25	0.1662
Final body weight (kg)	58.02 ± 2.04	59.79 ± 2.16	0.5616
Total gain (kg)	18.74 ± 1.08	17.62 ± 1.14	0.4849
Average daily gain (g/d)	193.13 ± 11.10	181.64 ± 11.79	0.4849
Total dry matter intake (kg)	115.96 ± 12.09	122.99 ± 12.37	0.7180
Feed conversion ratio ^3^	6.19 ± 0.35	7.12 ± 0.24	0.1281
Feed cost per kg weight gain ^4^	77.66 ± 4.45	86.52 ± 4.43	-

^1^ Experiment started when goats were 13 weeks old; experiment ended when goats were 37 weeks old; Total feeding days is 169 days. ^2^ Experiment started when goats were 37 weeks old; experiment ended when goats were 51 weeks old; Total feeding days is 97 days. ^3^ Feed conversion ratio is dry matter intake (kg) divided by body weight gain(kg). ^4^ Feed cost per kg weight gain is NT$ divided by body weight gain (kg).

**Table 3 animals-12-03516-t003:** Effects of castration at different ages on *longissimus dorsi* muscle in Nubian crossbred male goats at the end of the growing and fattening period (LSM ± SE).

Items	Castration at 3-Month-Old (*n* = 9)	Castration at 6-Month-Old (*n* = 8)	*p*-Value
Growing period end ultrasonic measurement ^1^			
*Longissimus dorsi* muscle area (cm^2^)	8.51 ± 0.28	9.36 ± 0.30	0.0551
*Longissimus dorsi* muscle depth (cm)	2.17 ± 0.06 ^b^	2.47 ± 0.06 ^a^	0.0032
*Longissimus dorsi* muscle width (cm)	5.06 ± 0.09	5.13 ± 0.09	0.6201
Backfat thickness (mm)	0.42 ± 0.05	0.55 ± 0.05	0.0962
Fattening period end ultrasonic measurement ^2^			
*Longissimus dorsi* muscle area (cm^2^)	13.01 ± 0.46	12.81 ± 0.49	0.7735
*Longissimus dorsi* muscle depth (cm)	2.86 ± 0.06	2.74 ± 0.06	0.1491
*Longissimus dorsi* muscle width (cm)	5.75 ± 0.07	5.69 ± 0.08	0.6205
Backfat thickness (mm)	0.52 ± 0.06	0.59 ± 0.06	0.4576

^a, b^ Means within the same row without same letter on superscripts are significantly different (*p* < 0.05). ^1^ Growing period: around 40 kg. ^2^ Fattening period: around 60 kg.

**Table 6 animals-12-03516-t006:** The economic analysis of castration at different ages for Nubian crossbred male goats.

Items	Castration at 3-Month-Old	Castration at 6-Month-Old
Experimental days in feedlot ^1^	266	266
Experimental feeding cost (NT$) ^2^	3329	3537
Experimental net earnings (NT$) ^3^	2908	3358

^1^ Experimental days in feedlot: Nubian × Boer crossbred male goats from 15.2 to 58.9 kg body weight. ^2^ Experimental feeding cost: [(Final body weight - Initial body weight in growing period) × feed cost per kg weight gain in growing period] + [(Final body weight − Initial body weight in fattening period) × feed cost per kg weight gain in fattening period]. ^3^ Experimental net earnings = Average auction price in December 2018 [33] (Boer × Nubian crossbred wether NT$ 254/kg) − (Experiment feeding cost + Cost of weaning kid); 2.5-month-old Nubian × Boer crossbred kid was NT$ 8500.

## Data Availability

The data are contained within the article.

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
