# Peer review of "Effects of Castration Age on the Growth Performance of Nubian Crossbred Male Goats"

_animals, 2022, doi:10.3390/ani12243516_

Round 1
Reviewer 1 Report
The article is well structured in the design of the study and in the clarity of the tables. The English language is neat and clear. I have a few considerations that I hope will provide insights to improve the quality of the article.
Dear authors, In the meantime, thank you for considering me as a reviewer for this article. Indeed, the manuscript has several points to its advantage. The study design is correct. The form and the English language are accurate, which shows the care and precision with which the authors worked. The tables are very clear. In summary, it is a good paper. After a few revisions, the article is in my opinion publishable in a journal. My only concern is about the originality and interest that the article might arouse in readers. I found the manuscript not very original and of interesting scientific impact. Despite the fact that it is quite narrow in terms of pages and lines, the discussions are a bit scattered and in my opinion can become burdensome for the reader to finish reading. In any case, I reiterate my positive assessment of the care and clarity of the drafting of the manuscript. I hope I have been helpful. Thank you again and I wish you best regards.
Author Response
Point 1: My only concern is about the originality and interest that the article might arouse in readers. I found the manuscript not very original and of interesting scientific impact. Despite the fact that it is quite narrow in terms of pages and lines, the discussions are a bit scattered and in my opinion can become burdensome for the reader to finish reading.
Response 1:
- At the end of the fattening period, the value of TG was in the normal range, consistent with the biochemical values of goats described by Bogin [26].
- The only significant difference was found in TES concentration (Table 4) since the cas-tration was done in the non-breeding season and the goats had not reached puberty, so the TES concentration was low, which was consistent with the result from Souza et al. [29] and Theron [30].
- The discussion section was revised according to the reviews, please refer to the attachment for more details. The revision parts were highlighted with yellow mark.
Thank you so much~
Reviewer 2 Report
The present manuscript aims to study the optimus timing for performing castration in male goats. The entire document is well referenced and structured. The contents are expressed in such a way that the reader will find the paper easier to read and comprehend.
I have only two minor suggestions, concerning grammar errors:
- Simple Summary: Line 10: I guess you would say 'to perform castration has not been scientifically determined'.
- Abstract: Line 29: Please correct maybe as 'may be'.
My sincere congratulations to the authors.
Author Response
Point 1: Simple Summary: Line 10: I guess you would say 'to perform castration has not been scientifically determined'.
Response 1: Castration is a common management procedure to improve meat quality in goats, but the optimal timing to perform castration has to be scientifically determined.
Point 2: Abstract: Line 29: Please correct maybe as 'may be'
Response 2:Castrating goats at 3-month of age may be a better practice as animal welfare and possible risks associated with late castration are taken into consideration.
Please refer to the attachment for more details, the revision parts were highlighted with yellow mark.
Thank you very much~~
Reviewer 3 Report
Please improve the material and methods as specified in manuscript.
Section 3.4, Line 253 to 256, the authors contention is enhancement in net earning as feeding cost and feed cost per kg weight gain is lower in goat castrated at 3 months of age, however, this is not well supported by data.
Similarly, section 3.4, Line 256 to 259, the authors have sold the animals since they could not maintain it for longer period due to factors not associated with experiment.
Hence, the analysis for economic benefits accruing to the commercial breeder needs to be refined, re-thought and re-framed.

Author Response
Point 1: Section 3.4, Line 253 to 256, the authors contention is enhancement in net earning as feeding cost and feed cost per kg weight gain is lower in goat castrated at 3 months of age, however, this is not well supported by data. Similarly, section 3.4, Line 256 to 259, the authors have sold the animals since they could not maintain it for longer period due to factors not associated with experiment.
Response 1:
The economic analysis of castration at different ages in Nubian crossbred male goats was shown in Table 6. Although the experimental feeding cost of 3-month castration group was lower than the 6-month castration group (NT$ 3329 vs. NT$ 3537), the higher BW (58.02 vs. 59.79) generate higher net earnings for the 6-month castrated group than the 3-month castrated group (NT$ 2908 vs. NT$ 3358). If the experimental duration was prolonged till the goats reached the major marketing body weight (70 kg), the estimated cost would be feed cost per kg weight gain in fattening period (77.66 vs. 86.52, Table 2) multiplying the weight gain from actual marketing body weight (58.02 vs. 59.79) to 70 kg, so the feed cost of 3-month group would be NT$ 161 lower than that of 6-month group (NT$ 4259 vs. NT$ 4420), and the net earnings would be higher (NT$ 5021 vs. NT$ 4860). Moreover, the risk of failure of castration could be withstand and the loss would be lower at 3-month castration group. Therefore, for commercial breeders, the market factors according to their breeds, management method, growth, fattening, to marketing should be considered to maximize the profits.
The discussion section was revised according to the reviews, please refer to the attachment for more details. The revision parts were highlighted with yellow mark.